# Mechanized Grape Harvest Efficiency

**Ján Jobbágy [1], Martin Dočkalík [1], Koloman Krištof [1,\*] and Patrik Burg [2]**

1 Department of Machines and Production Biosystems, Faculty of Engineering, Slovak University of Agriculture in Nitra, Hlinku 2, 94976 Nitra, Slovakia; jan.jobbagy@uniag.sk (J.J.); xdockalik@uniag.sk (M.D.)

2 Department of Horticultural Machinery, Faculty of Horticulture, Mendel University in Brno, Valtická 337, 69144 Lednice, Czech Republic; patrik.burg@mendelu.cz

\* Correspondence: koloman.kristof@uniag.sk

**Abstract:** Due to the low number of employees and the time limit in the field of grape harvesting, we focused in the presented article on evaluating the effectiveness of the deployment of an outboard grape harvester within the conditions of Slovak viticulture. The vineyards are in the Nitra wine-growing region on the southwestern slopes behind the Pivnica Radošina company, Piešťany district, in a total area of 33 ha. The object of the research was a PELLENC 8090 Selective Process trailed grape harvester aggregated with a SAME Frutteto3 100 tractor. The mechanized harvesting was tested in the vineyards of Pivnica Radošina s. r. on three selected varieties (Pinot Gris, Pinot Blanc, and Tramin Red) for two years. Entry conditions, technical parameters of the equipment, and weather conditions were monitored for all varieties. Data were analyzed with STATISTICA statistical software. As part of the research, post-harvest losses due to mechanized harvesting were monitored. The average losses for all examined varieties reached the value of 2.17% in 2018, and the value of 2.25% in 2017. A significant output was the efficiency of the deployment of the set for mechanized grape harvesting, where in 2018 a minimum value of 146.3 ha was set (the average cost of grapes was 500 Euro·t$^{-1}$). A significant part of the cost was fixed items, as a two-year tractor and a trailed collector (for 2017) were used for mechanized harvesting. The difference compared to previous research was the deployment of a completely new set during a two-year period on varieties that have not yet been evaluated. The economic efficiency over two years was elaborated in detail, which highlighted the benefits of multi-annual use. In terms of examining losses, differences were shown not only between varieties, but also between years, and these data were statistically verified. The paper evaluates the dependence of the use of mechanized harvesting on changes in the purchase price of grapes (increasing it also exponentially increases the required area) and on changing the hourly wage of an employee (increasing it degressively reduces the required area). From the results it can be said that statistically and economically significant outputs were achieved for the deployment of machine collection.

**Keywords:** grape; harvest; mechanization; losses

## 1. Introduction

On a nationwide scale, the vineyard is grown on an area of approx. 8 mil. ha, although within the Slovak Republic, it is currently grown on an area of 18,000 ha. Specific cultivation technologies are still characterized by a relatively high need for manual labor, which is 400 to 600 h·ha$^{-1}$ of fertile vineyard. One of the most demanding work operations in terms of labor is the collection of grapes, which accounts for up to 30% of the total need for working time (in practice it represents 120–160 h·ha$^{-1}$). Grape harvesting can be affected by several factors under operating conditions (meteorological factors, grape health, varieties, yield, type of line and planting clips, total area harvested). In recent years, the more complex availability of labor and the rising cost of human labor have been a significant factor. These aspects therefore lead to the increasing use of fully mechanized grape harvesting with the use of tractor-mounted or self-propelled harvesters [1]. In Slovakia, agriculture

records the most significant decline in employment of all sectors of the national economy. Back in 1995, agriculture was the third largest sector with almost 200,000 employees. At present, it belongs to the least numerous branch in the Slovak Republic, with the number of employees just exceeding 50,000 [2]. In the conditions of the Slovak Republic, the issue of the method of grape harvest and its economic evaluation is generally only studied by the research team from the university (SPU in Nitra). There is no such knowledge. Partially mechanized harvesting is so far the most widespread work procedure applied in the grape harvest. It is implemented mainly by small and medium-sized enterprises with growing areas, which are often scattered over several territorial units. Large companies use it mainly for over-harvest harvesting and for harvesting grapes intended to produce high-quality wines. In the 1990s, winegrowing enterprises in the Slovak Republic solved the problems associated with transformation and privatization and had a relatively sufficient supply of cheap labor at their disposal [3]. Therefore, most investments were initially directed to the provision of cultivation technologies, the renewal and restructuring of vineyards, and to processing technologies. However, it can be expected that the use of partially mechanized harvesting for large areas of vineyards will be very problematic in the future due to the permanent loss of seasonal workers, especially in terms of ensuring harvesting on schedule. The gradual loss of labor over the last 90 years and the ever-increasing price of manual labor on the market forced growers to invest funds in the purchase of modern viticultural equipment to maintain the profitability of production. This moment was also the impetus for the first wave of grape harvester use in operational practice in the years 2000 to 2006, realized mainly by transformed agricultural cooperatives or private entities with large growing areas [4]. Even in the former Czechoslovakia in the 1970s, experiments with mechanized collection took place, as stated by Otáhal [5], but at that time this method was not adopted and no results from the monitoring of losses from this period were published. The influence of mechanized harvesting on the quality of wine in terms of sensory properties in comparison with manual harvesting was monitored. These measurements were performed by the Comprehensive Research Institute of Viticulture and Enology Bratislava [6].

Thanks to the gradual establishment of modern grape harvesters in South Moravia, in the years 2006 to 2008 there was a demand for these machines (especially for semi-trailer machines) from medium-sized companies with an area of over 30 ha. At the same time, in these years, the first entities offering the possibility of mechanized collection in the form of a service also appeared on the market. The relevant shortage of grapes on the market in 2010 and the consequent effort to be as independent as possible from the supply of grapes from domestic growers led large wine producers to purchase large areas of cultivation and to gradually restructure them. This fact caused in the following years an increased demand, especially for multifunctional carriers enabling the performance of, in addition to fully mechanized collection, other operations, especially chemical protection. Over the course of ten years, the share of fully mechanized grape harvest in the total volume of harvesting operations in viticulture has increased several times in the Slovak Republic. The course of costs depending on the annual deployment of collectors in confrontation with the costs of collection provided in the form of services (346.64 Euro·ha$^{-1}$) and the costs of partially mechanized collection using collection trays (500.71 Euro·ha$^{-1}$) allow a closer look at the economic efficiency of monitored collectors [4]. The authors [7] state the price of partially mechanized harvesting in the conditions of Germany in the amount of 900 to 1200 Euro·ha$^{-1}$, the price of fully mechanized harvesting at the level of 450–600 Euro·ha$^{-1}$. For the self-propelled collector Gregoire G152, the effective area of annual deployment is in the range of 99–102 ha·year$^{-1}$, depending on the achieved performance of 0.36–0.44 ha·h$^{-1}$. The efficient use of the self-propelled collector New Holland VL6060 corresponds to the collection of an area of 120–124 ha·year$^{-1}$. The values are influenced by the achieved performance of 0.40–0.48 ha·h$^{-1}$. The effective use of self-propelled grape harvesters is significantly influenced by the possibilities of their use during the season as portal carriers of tools, especially in chemical protection, pruning of plants, pruning of vines, or in other

operations. For comparison, Walg [8] states an effective planting area of 50 to 53 ha·year$^{-1}$ for multifunctional carriers for the conditions of Germany, while for Chile the conditions Troncoso and Riquelme [9] state an effective use of 140 ha·year$^{-1}$. Authors [4] for the conditions of the Czech Republic state an area of at least 100 ha·year$^{-1}$.

Even after the establishment of the independent Slovak Republic, mechanized harvesting began to develop only very slowly and distrust in mechanized harvesting has persisted among grape growers until today. This is also one of the reasons why mechanized grape harvesting has been intensively monitored in the last 10 years. In addition to monitoring losses during mechanized collection, research work is also focused on manual collection. On a scientific basis, it is an attempt to prove the results and refute long-standing conclusions about high losses in mechanized collection. The article is therefore focused on the evaluation of the mechanized harvesting of grapes from a technical point of view by monitoring the total losses and economic efficiency of its deployment.

The cultivation of grapes and the registration of winegrowers in our conditions is subject to the so-called Viticultural Register, which is a register of winegrowers, winemakers, traders, and retailers of wine products. It contains data on vineyard areas, winegrowers, winemakers, traders, and retailers of wine products [10].

In Slovakia, at this time the cultivation of must and the wine production sector are facing problems of cheap competition. These problems can be solved by higher state support, as winegrowers and winemakers point to low support, but these problems should also be solved by optimizing production processes and reducing production costs, as practice shows that not all processes are well managed. Therefore, it is necessary to introduce more mechanization and thus replace the increasingly expensive cost and lack of human labor [11].

In our country, manual harvesting of grapes still prevails, and mechanized harvesting is only slowly expanding. This is caused by high procurement costs of technology, as well as the poor state of wine support settings in the Slovak Republic, when vine growers solve the dilemma of whether it still pays to grow vines at all and not invest in expensive equipment, as there is a problem with grape sales on the market [12].

On the other hand, in terms of the sustainability of the future of viticulture in Slovakia, and thus the ability to produce vineyards and increase the efficiency of vineyard cultivation, it is necessary to introduce the maximum amount of mechanization of work in the vineyard [11,12].

The introduction of mechanization and modernization in the field of agriculture and horticulture is today an integral part of the present. According to some authors, the primary factor for introducing mechanization in the field of grape harvesting and processing is the reduction of human labor and the reduction of annual costs, even if the initial costs are high [3,13].

To evaluate the quality of work, standard-supported methods are used, which assess the collection performance, total losses in mechanized collection, and several other indicators. In the application of any of the methodologies, which we will deal with in this article, it is necessary to use mechanized collection with experienced staff [12].

As other researchers [14] stated, the current economic conditions force larger winegrowing enterprises over 40 ha to look for ways to make production more efficient. In addition to the purchase price and operational reliability, users and especially potential users are also interested in post-harvest losses.

In recent years, mechanized grape harvesting and processing has also advanced in Slovak viticulture. From the original three grape harvesters, their number has increased to more than 15 in the last 15 years. On the other hand, the prejudices of some older and especially smaller winegrowers are quite negative against the mechanization of harvesting. Over the last decade, the qualitative level of the mechanization of grape harvesting has clearly increased, which we will demonstrate in the article based on the results achieved [15,16].

Therefore, the main goal of this paper was to evaluate the effectiveness of mechanized harvesting in the form of post-harvest losses and the economics of using a grape harvester directly in the conditions of Slovak viticulture.

## 2. Materials and Methods

### 2.1. Terrain Conditions

The research was focused on selected vineyards of the company Pivnica Radošina s. r. o situated in the district of Piešťany (Slovak Republic). The Nitra wine-growing region is in the western part of the country and the vineyards are situated behind the company Pivnica Radošina (GPS coordinates: 48.55449 N 17.92922 E), oriented south. They reach a slope of 3–7% and fall under the wine-growing district Radošina (Figure 1).

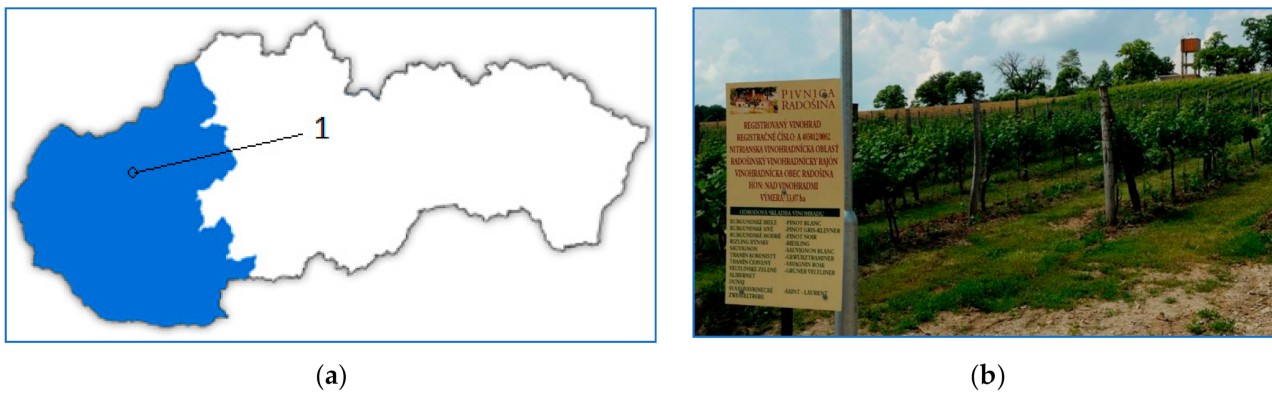

(**a**)  (**b**)

**Figure 1.** Map of the area of interest, with 1 denoting the location of the vineyards (right, cellar, 2020): (**a**) Description of study area with specific region highlights; (**b**) Detailed picture of study area of vineyards.

The long-term average of total atmospheric precipitation at a selected precipitation measuring station in Slovakia (Piešťany Weather Station) for the period 1989–2018 was 563.6 mm. The year 2017 was very warm in Slovakia (especially in western Slovakia) and in terms of the amount of atmospheric precipitation was within normal (524 mm). In Slovakia in 2017, the territorial annual average rainfall reached 827 mm. The average territorial air temperature in Slovakia was 8.4 °C, while the deviation from the normal in 1961–1990 was +1.3 °C. The total atmospheric precipitation at a selected precipitation measuring station near the site of interest was at the level of 499.7 mm in 2018 (representing 88.7% of the long-term average). At most selected stations, the total precipitation for 2018 was lower compared to the long-term average for the period 1989–2018. The average territorial air temperature in Slovakia was 8.6 °C. The altitude of the weather station for data collection was 163 a.s.l [17].

The southern orientation of the vineyards and the limestone–dolomitic subsoil make the Burgundy vine varieties, which have been grown here since time immemorial, stand out. Thanks to the deep soils with the already mentioned limestone–dolomite subsoil, the wines have a unique, unmistakable taste. The planting of the vineyard consists of originally grown traditional varieties Pinot Gris, Pinot Blanc, Pinot Noir, Sauvignon, Tramin, Riesling, Zweigeltrebe, Svätovavrinecké, Alibernet. They have also planted varieties from the ranks of Slovak new nobles in the given company, namely Danube, Nitra, Rosa, and Hetera. The total area of vineyards of interest is 33 ha.

In a winery of interest, we focused on the evaluation of mechanized harvesting of three different varieties of Vitis Vinifera Pinot Gris, Pinot Blanc, and Tramin Red. The characteristic input parameters of the vineyards are given in Table 1.

**Table 1.** Characteristics of vineyards.

| Cultivar | Spacing | Average Harvest, t·ha$^{-1}$ | Surface, ha | Number of Rootstock, Individuals·ha$^{-1}$ | Trellis System | Field Slope, % | Plant Age, Years | Trunk Height, m |
|----------|---------|------------------------------|-------------|---------------------------------------------|----------------|----------------|------------------|-----------------|
| Pinot Gris | 2.4 × 1.0 | 10 | 9 | 4200 | Double Guyot | 3 | 2011 | 0.8 |
| Pinot Noir | 2.4 × 1.0 | 12 | 2 | 4200 | Double Guyot | 3 | 2011 | 0.8 |
| Tramin red | 2.4 × 1.0 | 7–9 | 5 | 4200 | Double Guyot | 4 | 2011 | 0.8 |

Pinot Gris variety. It comes from France, where it was widespread, especially in the Champagne region, and reached the countries of Germany, Switzerland, and Central European. In our country, it is grown on approximately 100 ha, which represents 0.66% of the planting area of the vineyard. It has been registered since 1941. Frost resistance is good; similarly, it has good drought resistance. It springs moderately early, and blooms in the first decade of June. It has medium to high demands on position. The warm southern and southwestern slopes are very suitable. It likes deep and nutritious soils. Clay soils are very suitable, on which a larger number of extractives is formed. The variety can be grown on medium and high lines with a cut on the pull. In the vineyard in the company Pivnica Radošina, it was planted on an area of 9 ha (4200 individuals·ha$^{-1}$) [18].

Pinot Noir variety. Pinot Noir, otherwise known as Burgundy White, has its origins in French viticulture. This variety undoubtedly belongs to our vineyards and if it meets its growing requirements, the quality of wines will satisfy even the most demanding growers and consumers. Wines from this variety have a neutral taste. Pinot Noir ripens in the middle period and in most vintages, there are usually no problems with ripening grapes. This variety requires the best locations. Pinot Noir is also demanding on soils that are to be fertile and sufficiently supplied with moisture. The variety is more suitable for wider clamps than for the middle line, due to the intensive growth in deeper soils, when the fringes may shower. In the vineyard in the company Pivnica Radošina, it was planted on an area of 2 ha (4200 individuals·ha$^{-1}$) [18].

Tramin red variety. It belongs to the oldest cultural varieties of vineyards. It was probably created by crossing with forest vines and participated in the creation of other European varieties. Tramin red is an aromatic white variety, widespread in cultivation in Slovakia, the Czech Republic, but also in other countries. It is of medium growth, has dense foliage and demands intensive treatment of the leaf area. It is characterized by good frost resistance, but crops tend to be very volatile. Its demands on the soil are high. It thrives well in deep, fertile, sufficiently moist but warm soils. In the vineyard in Pivnica Radošina, it was planted in 2011 on an area of 5 ha (4200 individuals·ha$^{-1}$) [18].

The vineyard is grown here on a middle line with shaping into a flat harvest. The supporting structure consists of galvanized steel posts, suitable for mechanized collection.

### 2.2. Mechanization for Grape Harvest

The PELLENC 8090 SP trailer harvester kit (Pellenc, Quartier Notre Dame, Pertuis-France, Table 2) was used for aggregation with the SAME Frutteto3 100 vineyard tractor (SAME, Treviglio, Italy, Table 3, Figure 2). The Pellenc 8090 SP grape harvester (Table 2) is suitable for small and medium-sized enterprises. It is an over-the-counter grape picker that achieves lower performance compared to self-propelled machines. The combine is connected to the tractor by means of a lower link via special swivel drawbars, which allows good maneuverability of the set. The construction of the machine is portal. The machine works on the vibrating principle of shaking. The collecting device consists of bilaterally mounted pairs of shaking rods anchored at two points. These are rods of elongated shape. The catch device is lamellar with an inclination on both sides. Each side of the catch ramp

contains 19 slats, which are attached to flexible joints. The volume of the tanks is 3000 L. The control of the machine and the setting of working parameters is performed by means of a control unit consisting of a screen for displaying information, a combined joystick controller and a controller for navigating and setting parameters. The working speed of the set was in the range of 2.5–3.1 km·h$^{-1}$.

**Table 2.** Technical parameters of trailed grape harvester Pellenc 8090 SP [19].

| Parameter | Value |
|---|---|
| Hopper height when emptying, mm | 2300–2900 |
| Length/Width/Height, mm | 4670–4900/2550/3280–3880 |
| Minimum width of the intermediate row, mm | 1500 |
| Tank capacity, m$^3$ | 3 |
| Lateral inclination, % | 27 |
| Availability, % | 25 |
| Year of production | 2016 |

**Table 3.** Technical parameters of tractor Same Frutteto3 100 [20].

| Parameter | Value |
|---|---|
| Cylinders/Displacement, n/cm$^3$ | 4/4000 |
| Max homologated power, kW | 71 |
| Nominal engine speed, rpm | 2200 |
| Dimension, Length/Width/Height, mm | 3368/1618–2018/2314 |
| Year of production | 2016 |

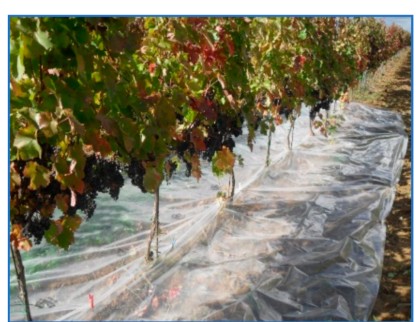

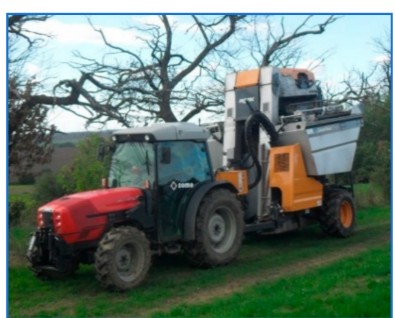

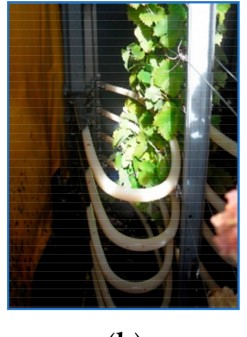

(**a**)　　　　　　　　　　　　　　　　　　　　　　(**b**)

**Figure 2.** Field experiments and equipment used. (**a**)Trial plots; (**b**) Tractor Same Frutteto3 100, trailed grape harvester Pellenc8090 SP and detailed picture of thresher mechanism.

*2.3. Combine Harvester Performance*

In terms of the deployment of tassels, the values for all varieties were in accordance with the requirements for mechanized harvesting, ranging from 0.7 to 1.3 m. As already mentioned, the working device consists of 10 rods (concept 2 × 5). The frequency of rod oscillations was for individual varieties from 470 to 490 min$^{-1}$. The oscillation amplitude was set to 100% and the throughput gap to 20 mm. The fan speed, for all three varieties, was set at 1600 rpm. For all three varieties, the working speed was in the range of 2.5–3.1 km·h$^{-1}$. These values were set according to the varieties and according to the experience of the combine harvester operator. The cleaning and sorting device is thus equipped with a set of fans and sorting flexible conveyors on both sides of the machine. In the upper part, there is an additional separation of the berries from the bunches, which could tear or break during harvest. The filling level of the hoppers is shown on the display of the control panel (in%), the hoppers are emptied by tilting them back hydraulically. The effective field capacity $C_a$ [21] was evaluated by measuring the working time during the experiments (Formula (1)).

$$C_a = 0.1 \cdot s \cdot w \cdot E_f; \; \left( \text{ha·h}^{-1} \right) \tag{1}$$

where:

$C_a$—area capacity, ha·h$^{-1}$,
$s$—field speed, km h$^{-1}$
$w$—distance between rows, m,
$E_f$—field efficiency, considering the time required for turning and maneuvering at the ends of the field and for hopper unloading [12,13,22].

The results are evaluated as mean with deviations (±SE), where input data such as working time and handling time were recorded and evaluated for the indicated varieties.

2.3.1. Field Trials

The research was carried out in a series of three measurements for each variety, i.e., a total of 9 measurements. In selected experimental rows (in the middle part of the row, so that the speed and height adjustment, the given variety, do not affect the correct research), PE foils were distributed in the ankle strip in a section 10 m wide, with 2.8 m and 1.4 m on each side of the row vines (Figure 2). Thus, one measurement represented one 10 m measured section. After passing the collector through the measured section, the fallen berries were collected in containers and then the berries were collected from the trusses that remained after passing the collector (losses of two kinds: losses by falling to the ground and losses of remaining tassels on steps) [23]. The total losses per 1 ha of vineyard were determined by weighing and calculation and expressed as a percentage of the harvest. The Statistica 10 program (StatSoft, Inc., Tulsa, OK, USA) was used to statistically evaluate the results and look for significant differences.

2.3.2. Efficiency of Using the Grape Harvesting Kit

Machinery used in agriculture is characterized by a group of properties, which we call technical–operational. Based on the definition of a group of technical and operational characteristics for a specific agricultural machine, its use in certain production conditions can be considered. "Use of the machine" means how these characteristics can be used in operating conditions in relation to the efficiency of deployment. We will consider the use of the machine to be economically efficient which, even with its increased purchase price, will not reduce the economic efficiency of the production process (Table 4). The efficiency of the use of increasingly expensive machines will depend primarily on the level of the technical and economic thinking of production managers [24].

For the correct deployment of the machine, it is necessary to know the cost structures in relation to the purchase price of the machine—operation of the machine with emphasis on the share of fixed and variable costs. In addition, it is necessary to get acquainted with technical and economic risks associated with the operation of machines and to know the

methods of calculating the most important criteria of economic efficiency of machines and lines (calculation of economic efficiency, profit, profitability, return on capital spent on machine purchase, etc.). We have introduced the term "limit of economic efficiency" for practical reasons, and that is the evaluation of the efficiency of the operation of technology at the company level. An important prerequisite for such an evaluation is the knowledge of the so-called unit operating (direct) costs and the possibility of comparing them with the costs of other technology operators, e.g., with the costs of service enterprises [25].

**Table 4.** Machinery cost parameters.

| Parameter | Units | Trailed Harvester | Tractor |
|---|---|---|---|
| Purchase price | Euro | 165,000 | 64,000 |
| Estimated life | h | 3000 | 10,000 |
| Annual use | h | 68.75 | 800 |
| Depreciation | year | 4 | 4 |
| Insurance and housing | Euro | 0 | 55 |
| Machine dimensions | mm × mm | 2500 × 4670 | 2020 × 3370 |
| Foreign capital | Euro | 148,500 | 57,600 |
| Own resources | Euro | 16,500 | 6400 |

When using a set of trailed grape harvesters in combination with a tractor, we also monitored the economic side of the method of harvesting in comparison with manual harvesting. The overall evaluation will follow the methodological procedures for determining the total costs (fixed, variable, and indirect costs). Fixed costs will include depreciation, interest, road tax, insurance, and garage [21,24,26]. When calculating costs, information concerning insurance and tax was obtained from the offer of insurers and compliance with the rules according to the prescribed laws [27].

The total direct annual cost of harvesting the grapes shall be calculated as:

$$rNmC = rNmT + rNmZ + rNo + rNe + rNzp; (Euro \cdot year^{-1}) \qquad (2)$$

where:

rNmT—direct annual costs of the tractor, Euro·year$^{-1}$;

rNmZ—direct annual cost per collector, Euro·year$^{-1}$;

rNe—annual energy costs, Euro·year$^{-1}$;

rNo—annual repair costs, Euro·year$^{-1}$;

rNzp—annual cost of living work, Euro·year$^{-1}$.

To correctly determine the total costs, it is also necessary to consider the costs of road tax [28]. Due to the agricultural tractor and grape harvester used, the cost of the road tax was zero. According to the valid legislation, the tractor was insured by compulsory statutory insurance, where the insurance costs amounted to 55 Euro (for 2017 and 2018). Annual garage costs are based on the dimensions of the machine and the cost per 1 m$^2$ of garage space per year (own space-0 Euro·year$^{-1}$). Depreciation costs are part of the rNmT (direct annual costs of the tractor) cost for the tractor and the rNmZ (direct annual cost per collector) for the collector.

The annual variable costs therefore consist of the costs of repairs, energy, and live work. When solving the calculation of the costs of the set, we proceeded according to [25]. When calculating the costs of the set, it is necessary to count the use of the energy means only according to the respective share of use. To evaluate the efficiency of the use of the combine for the harvesting of grapes, the results obtained by us in the harvesting of different varieties were considered and, on the other hand, these are the costs that would be incurred in solving the harvesting services. The economic evaluation was also extended

to the assessment of the necessary minimum area of vineyards. The total cost of manual work (contract work) was 5.4 Euro·h$^{-1}$ (with levies for the employer). Based on our survey, the time required for collection was considered to be 102 h·ha$^{-1}$. As stated by [25], the values of costs are also given in the category of annual expression (Euro·year$^{-1}$) in relation to the use of agricultural machinery. To assess the effective deployment of technology, one of the important indicators is the currently analyzed costs expressed in relation to units of measurement (Euro·ha$^{-1}$, Euro·h$^{-1}$, Euro·t$^{-1}$). Both categories are to be considered as variables in the time function $f(t)$.

Zero losses were considered for manual harvesting, as losses are usually very low for manual harvesting [29].

### 2.3.3. The Annual Variable Costs Therefore Consist of the Costs of Repairs, Energy and Live Work

Annual repair costs

The annual repair costs are determined according to the ASABE standard [21], where the correction factors according to the standard, the year of manufacture of the machine, and the annual use of the machine are applied for the calculation. Regarding the year of acquisition of the tractor and collector, we must also consider the inflation rate in the calculations (harvester, average for the period from 2002 to 2014, $i = 0.04$; tractor, average for the period from 2004 to 2014, $i = 0.03$).

$$_rN_o = RF_1 \cdot C_s \cdot (1+i)^n \cdot \left(\frac{h}{1000}\right)^{RF_2}; \qquad (\text{Euro·year}^{-1}) \qquad (3)$$

where:

$RF_1$ and $RF_2$—correction coefficients (wide range according to the type and reliability of the machine, ASABE Standards);
$C_s$—acquisition (entry) price of the machine, Euro·year$^{-1}$;
$I$—average inflation;
$N$—age of equipment, year.

Annual energy costs

$$rNe = Q \cdot Ce \cdot rW \cdot 1.1 \qquad (\text{Euro·year}^{-1}) \qquad (4)$$

where:

$Q$—energy consumption, L·ha$^{-1}$;
$Ce$—energy price, Euro·L$^{-1}$;
$rW$—annual use of the machine, ha·year$^{-1}$.

Annual cost of live labor-tractor operator ($_rN_{zpm}$) for mechanized harvesting (the relationship applies if the work is performed by one worker):

$$_rN_{zp} = {}_hN_{zp} \cdot 1.352 \cdot \frac{_rW}{C_a}; \qquad (\text{Euro·year}^{-1}) \qquad (5)$$

where:

$_hN_{zp}$—hourly wage of the worker, Euro·h$^{-1}$;
$_rW$—annual use of the machine, ha·year$^{-1}$;
$C_a$—Effective field capacity, ha·h$^{-1}$.

The direct unit cost of $_jN_{mC}$ as part of the depreciation (use) period and strategy can then be determined from the:

$$_jN_{mC} = \frac{_rN_{mC}}{_rW}; \qquad (\text{Euro·year}^{-1}) \qquad (6)$$

where:

$_rN_{mC}$—direct annual costs, Euro·year$^{-1}$;

$_rW$—annual use of the machine, ha·year$^{-1}$.

The economic assessment was also extended to include an assessment of the necessary minimum area of vineyards when changing the cost of live work, ranging from 2 to 14 Euro·h$^{-1}$ including VAT. The total costs were also considered in the case of manual harvesting in the same vineyards, as the holding does not collect everything in a mechanized way. There are certain losses in manual harvesting, but we did not deal with them because they depend on many input factors (size of area, length of rows, variety and thus arrangement of grapes on vine bush, physical and psychological utilizing of a person, etc.). Based on our survey, the time need for harvesting in the area of 102 h·ha$^{-1}$ was considered.

## 3. Results and Discussion

### 3.1. Harvester Working Capacity

In view of the results obtained, it can be determined that statistical differences have been found regarding the difference in varieties. In terms of fertility assessment, the highest values are for the Ruland white variety (13.88 ± 0.42 t·ha$^{-1}$), medium for the Ruland grey variety (10.60 ± 0.35 t·ha$^{-1}$), and lowest for Tramin red (9.36 ± 0.28 t·ha$^{-1}$, (mean ± SE)).

The statistical evaluation of the results of the harvested grape varieties during 2017 and 2018 shows that the differences were not statistically significant, either between years ($p = 0.17$; *Fcrit* = 7.71; *F < Fcrit*) or between varieties ($p = 0.34$; *Fcrit* = 9.55; *F < Fcrit*).

In the recovery of sugar content with respect to the harvest date, the values for the Ruland grey variety were 21° NM, the Ruland white varieties 18.5° NM (collected per sect) and for Tramin red 22° NM.

Although 2018 was characterized by good conditions (varieties matured earlier), 2017 was more profitable based on the appreciation of the harvests achieved. It can therefore be said that in 2017 above-average harvests were achieved for the Tramin red varieties (13.496 t·ha$^{-1}$) and Ruland whites (16.374 t·ha$^{-1}$). In that year 2017, the same frequency of shake rods (450 min$^{-1}$) was set, the amplitude was set to 90%, the passability gap of the collecting system to 20 mm, and the speed of the cleaning fans to 1500 min$^{-1}$.

In 2018, after previous experience, the values of driving speed and frequency of rod oscillations were adjusted, depending on the thickening of the stand and the time of grape harvest (degree of ripening). Non-working time periods such as turning and emptying time are influenced by the type of technique and its flexibility and the experience of the operator, respectively given by terrain conditions. Practical experience shows that the frequency of oscillations is given by the term of grape harvest, where the berries are better kept on the bunch during earlier harvesting and the frequency needs to be increased. Other authors who document their results in scientific articles [13,30–33] met with identical results.

As part of the evaluation of the efficiency of the time use of working time for three varieties, the time difference created by the slope of the given terrain and the concentration of the operator was proved. The maximum difference with respect to the Pinot Gris variety reached a deviation value of up to 5%. The results show (Table 5) that the time periods for maneuvering and emptying represent an average of 180 s. Field efficiency $E_f$ reached a value in the range from 0.72 to 0.75.

The values of effective machine capacity (effective field capacity) showed differences in the maximum value of 11.1% compared to the Tramin red variety. The value of material field capacity depends primarily on the fertility of the given variety and on the performance, where significant differences were shown, namely the value of 2.45 t·h$^{-1}$ (Pinot Noir variety versus Tramin Red variety). Evaluation of the results of field capacity are presented in Table 5. The results again indicate the influence of the slope of the terrain and the experience of the operator; the maximum difference between the examined varieties 6.1% (compared to the Tramin Red variety). In our conditions, manual harvesting is carried out by a group of people who harvest the grapes mostly in pairs for each row of the vineyard.

**Table 5.** Operating characteristics of grape harvesters (mean $\pm$ SE).

| Characteristic | Units | Variety | | |
|:---:|:---:|:---:|:---:|:---:|
| | | cv. Ruland Grey | cv. Ruland White | cv. Tramin Red |
| Field speed | km·h$^{-1}$ | 2.8 $\pm$ 0.3 | 2.8 $\pm$ 0.3 | 2.8 $\pm$ 0.3 |
| Turning time | s | 62 $\pm$ 2.0 | 63 $\pm$ 2.2 | 65 $\pm$ 2.3 |
| Unloading time | s | 95 $\pm$ 1.7 | 96 $\pm$ 1.8 | 97 $\pm$ 2.0 |
| Beater frequency | beats·min$^{-1}$ | 490 | 470 | 490 |
| Field efficiency, $E_f$ | - | 0.75 | 0.74 | 0.72 |
| Effective field capacity, $C_a$ | ha·h$^{-1}$ | 0.50 | 0.48 | 0.45 |
| Material field capacity | t·h$^{-1}$ | 5.34 $\pm$ 0.19 | 6.66 $\pm$ 0.21 | 4.21 $\pm$ 0.17 |
| Field Capacity, (manual harvest) | $10^{-3}$·ha·h$^{-1}$ | 10.10 | 9.80 | 9.52 |

As mentioned, the total area of vineyards covers an area of 33 ha, where the three varieties examined make up 48.5% of the total area. In 2017, the working gear settings changed from the original 450 min$^{-1}$ for each variety to 490 min$^{-1}$ for the Pinot Gris and Tramin Red varieties, and to 470 min$^{-1}$ for the Pinot Noir variety. The amplitude of the oscillations has changed from the original 90% (2017 for all varieties) to 100%. The fan speed (1600 rpm) and the throughput gap (20 mm) remained unchanged. The travel speed was maintained at an average value of 2.8 km·h$^{-1}$ for all varieties. The value of effective field capacity $C_a$ in our research ranged from 0.45 to 0.5 ha·h$^{-1}$.

Within the research in the conditions of the Slovak Republic, the effective field capacity $C_a$ was also evaluated in the winery in Doľany in three examined varieties (Pinot Noir, Neronet, and Veltlin Green). The total area of cultivated examined varieties was 23.1 ha. The results show that for the given varieties the average value of 0.31 ha·h$^{-1}$ was achieved for the used Tractor Lamborghini + trailed harvester Ero LS Traction kit [12]. Approximate values (0.34 ha·h$^{-1}$) of the effective field capacity parameter were also achieved by other authors [13] in the cultivar cv. Trebbiano with a used combination consisting of a CNH T5060 tractor and a trailed harvester ERO LS Traction. In other measurements with a set consisting of a Zetor 7311 tractor and an ERO LS Traction trailed pick-up, a value of 0.24 ha·h$^{-1}$ was reached (harvested variety Frankovka) [14].

In comparison with our results, which we achieved in 2018 on the mentioned varieties (Table 5), the values of effective field capacity $C_a$ were higher for all varieties by at least 32%.

### 3.2. Harvest Losses

During the evaluation of the research, in addition to the evaluation of the harvester working capacity parameter, we also focused on the losses that occur not only during the mechanized collection, but also during the manual collection. However, in the case of manual harvesting, this value is clearly influenced by the persons working in the vineyard. However, in this article we focused on the recovery and losses arising from mechanized collection. The losses that occur here can be divided into losses by the fall of the grapes to the ground and losses consisting of unharvested grapes. The average losses for all three examined varieties reached the value of 2.17%, where 1.1% were losses of unharvested grapes and 1.07% were losses by falling to the ground. Given the influence of the variety on the losses incurred, we can talk about the demonstration of dependence, where for the Pinot Noir the losses averaged 3.23% (448.56 kg) and for other varieties up to 2% (Pinot Gris: 1.82% (193.06 kg) and Tramin red: 1.47% (137.34 kg)). From the point of view of the setting of the shaking device (number of oscillations) and the resistance of the variety of separation of berries from bunches, it can be said that the value of losses by non-separation of 2.4% was achieved for the Pinot Noir variety. Losses by falling to the ground represented

maximum values of, on average, 1.26% (evaluated year 2018). Compared to 2017, where the average losses for the Pinot Noir and Tramin Red varieties were 1.75%, we lost 0.42% of the total varieties on the examined varieties. This was mainly due to an increase in the losses of unseparated grapes from 0.62% to 2.4% for the Pinot Noir variety (Table 6). It is worth considering whether this caused an increase in the vibrations of the shaker from 450 to 470 min$^{-1}$, or the date of collection. Overall, however, mechanized collection proved to be effective in terms of loss assessment, as total losses were up to 5%. However, by comparing the varieties from the point of view of the setting of the shaking device, the reduction of the number of oscillations by 20 min$^{-1}$ reduced the losses of both other examined varieties.

**Table 6.** Harvest losses, % of production, cv. Ruland Grey, Ruland White, and Tramin Red (2018).

| Variety | Undetached Grape | Grapes on the Ground | Total |
|---|---|---|---|
| Ruland Grey | $0.56 \pm 0.01$ | $1.26 \pm 0.13$ | $1.82 \pm 0.14$ |
| Ruland White | $2.40 \pm 0.83$ | $0.83 \pm 0.07$ | $3.23 \pm 0.82$ |
| Tramin Red | $0.34 \pm 0.13$ | $1.12 \pm 0.05$ | $1.47 \pm 0.13$ |
| Average losses | 1.10 | 1.07 | 2.17 |

In addition to the average yields and losses achieved by non-harvesting or fall in individual grape varieties, we also focused on the statistical evaluation of the achieved results. From the point of view of statistical evaluation by one-factor analysis of Anova at the level of significance of 95%, it can be said that in 2017 the total losses of individual varieties did not show significant differences ($p = 0.39$; $p > 0.05$; *Fcrit* = 5.14; *F < Fcrit*). The results for 2017 therefore show that there is no statistically significant difference between the individual varieties examined in terms of total losses. However, the turning point came in 2018, when yields were lower on average. In a given year, the results indicated statistically significant results ($p = 0.0027$; $p < 0.05$; *Fcrit* = 4.07; *F > Fcrit*). In terms of statistical evaluation, we decided to subject the results obtained to the impact of the year of collection on the total losses incurred. From the results, it can be said that statistically significant differences were achieved for one variety (Tramin Red) ($p = 0.0025$; *Fcrit* = 4.07; *F > Fcrit*) significant differences ($p > 0.05$).

According to the results of authors [12,14,33] it can be stated that acceptable values have been achieved by standard and practice. According to older sources, i.e., [34], a loss rate of 3.0–12.0% is permissible.

When the grapes are harvested, losses occur which ultimately affect the overall harvest. In the case of mechanized harvesting, these are losses by falling to the ground and losses by non-harvesting. Scientific studies dealing with the issue point to results with values oscillating around 3% as appropriate. The number of losses depends on the input parameters of a technical or technological nature (travel speed during harvest, frequency and amplitude of oscillations, fan speed, and throughput gap), on the characteristics of the stand (stand density, ripeness, variety) and on weather and soil conditions. From a technological point of view, the ripeness of the grapes influences the setting of the frequency of the vibrations of the shaking device. In the case of mechanized harvesting of grapes, the operating parameters are set every year due to the change in the state of ripeness and variety during the first 20–30 m [12,30–33].

No losses were considered in the manual collection when evaluating the results. By increasing the area, in such a state, the costs of losing the grapes due to non-harvesting or falling are not created. Subsequent comparison thus puts mechanized collection at a disadvantage. However, we know from our own experience that with manual collection, especially in terms of time and physical demands of work, there are both losses by dropping and losses by abandonment. However, the research activity is more complicated because it depends on the selected specific line and thus on the collector that worked on the given line. Unfortunately, we do not have the results of manual collection and their losses available, and therefore we cannot quantify the findings, substantiate them with the

resulting numbers, and thus directly compare the losses incurred during mechanized and manual collection. Demonstrable and monitored losses by research are confirmed by the results [35], where they achieved losses by falling berries on the ground during manual harvesting up to 338 kg·ha$^{-1}$ (with a grape harvest of 12.8 t·ha$^{-1}$–2.64%).

*3.3. Efficiency of the Use of Mechanized Grape Harvesting*

As stated by [24], the main economic indicators, according to which the effectiveness of mechanization means is evaluated, are the direct costs of the monitored operation with the application of the given machine, respectively machine set; in our case, the trailed grape harvester and the tractor, and the quality indicators of the work of this machine (set) relating to the unit of performance.

We evaluated the cost of a mechanized harvesting machine kit that was used in our research measurements. Market prices of technology and interest rates during the period of our research activities were used as input variables; other values of the input parameters were obtained from the selected company, as well as values corresponding to the technique used [12,36].

The total use of the set for mechanized grape harvest (for an area of 33 ha) was 68.75 h. With regard to the total purchase price of the set (with regard to own resources and bank credit), the total use of the machine set for grape harvest, depreciation time, energy costs and repairs amounted to 53,946.11 Euro·year$^{-1}$ (year 2017, age of tractor and harvester 2 years) and 53,006.45 Euro·year$^{-1}$ (year 2018, age of tractor and combine harvester 3 years), which results in unit costs 1634.73 Euro·h$^{-1}$ (year 2017) and 1606.26 Euro·h$^{-1}$ (year 2018).

Based on available information from various companies offering services, we found that the market price of labor-mechanized grape harvesting using a trailed grape harvester ranges from 450 to 490 Euro·ha$^{-1}$ with VAT. Given the time-consuming nature of manual collection (in our conditions 102 h·ha$^{-1}$) and the cost of an hourly wage of an employee with contributions, it was found that the unit costs amount to 534.6 to 567 Euro·ha$^{-1}$. It clearly follows from these values that it is more advantageous to order mechanized harvesting of grapes (without considering losses in manual or mechanized harvesting). However, due to the nature and opinions of winemakers, the result is such that they often remain loyal to traditional manual harvesting, especially regarding entry conditions (vineyard area, vineyard location and access of machinery, types of posts and lines-affects the possibility of using equipment). The results of the machine collection show that this will not be enough when using the kit only for an area of 33 ha. As the company has only a total area of vineyards of 33 ha, it is recommended to use a collector in the form of services from the achieved results. This will increase the efficiency of the use of mechanized grape harvesting.

To determine the economic items of individual examined grape varieties, high unit constant costs were achieved for individual varieties, which are related to the low area of the given variety (also valid for 2017 and 2018, Figure 3). The constant costs were considered the same for the individual varieties as well as for the total area of the vineyard. The difference in the recalculation of constant costs between 2017 and 2018 was 28.65 Euro (a decrease of 1.8%). Due to harvested varieties and settings of harvesting parameters, indirect costs decreased by 30% in 2018 compared to 2017. The change in variable costs due to energy input prices, which changed slightly (fuel, oil, hourly wage), was 0.3% (between 2017 and 2018).

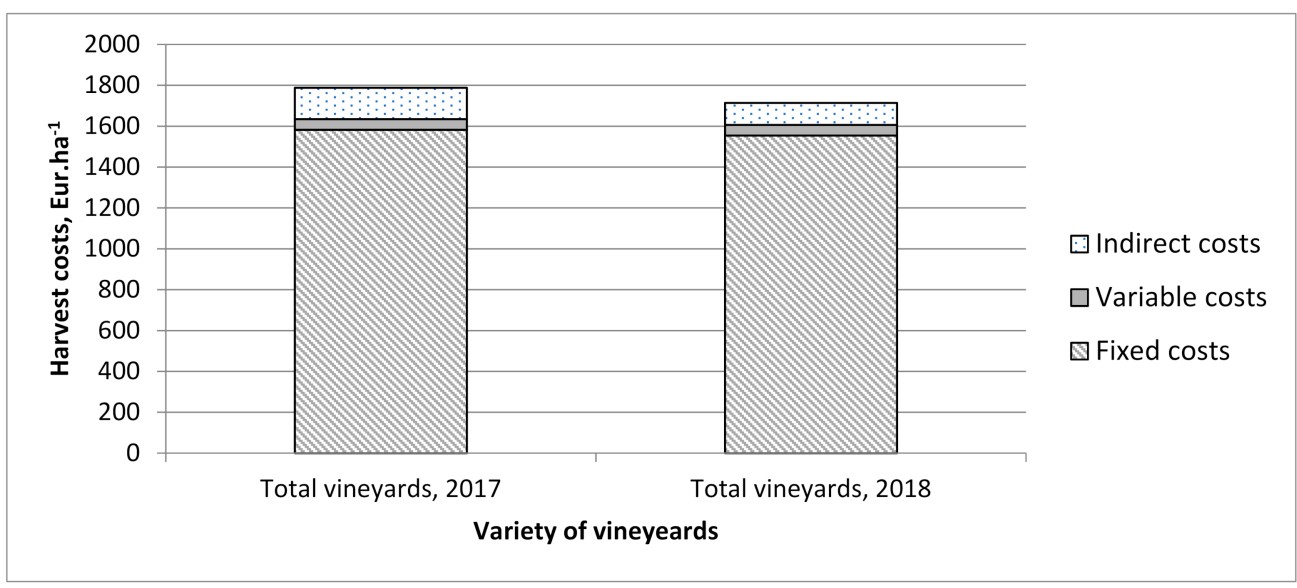

**Figure 3.** Harvesting costs (Fixed, variable, and indirect) for total vineyards, year 2017 and 2018.

When assessing the deployment of the set in the range from 10 ha to 200 ha, differences in the percentage range of 1.39% to 1.78% were achieved between 2017 and 2018. The efficiency of using the machine set for grape harvest reached the level of 104.76 ha (year 2017) and in 2018 the value of 102.9 ha. Indirect costs, which are formed by the loss of grapes, must also be considered when evaluating the effectiveness of the use of the grape harvester. In 2017, according to the examined varieties, these values amounted to indirect costs ranging from 119.98 Euro$\cdot$ha$^{-1}$ (Pinot Noir variety) to 202.65 Euro$\cdot$ha$^{-1}$ (Pinot Noir variety) and in 2018 the average indirect costs from examined varieties were 107.52 Euro$\cdot$ha$^{-1}$. Due to the identified losses for the year of the 2017 survey, the value of the required minimum area increased to 173.5 ha and in 2018 to 146.3 ha (considering average prices for grapes of 500 Euro$\cdot$t$^{-1}$). A significant difference is the parameters of the mechanized collection settings, which were made to reduce losses. By increasing the annual performance of the set, specifically the techniques for grape harvesting, lower operating costs are achieved reflected in the harvested area. When evaluating the total unit costs with respect to the total area of 33 ha, the fixed costs were up to 30.23 times higher than the variable costs. Amortization costs were not zero, as this is a new technique (up to 4 years). Road tax in our country does not yet apply to this technique. A significantly high item was made up of annual depreciation costs and capital interest on total fixed costs. In the evaluation of variable annual costs, the highest item was, for individual varieties and small areas, energy costs [29,36].

When evaluating the total unit costs in 2017 and 2018 by a graphical representation as seen in Figure 4, the level of degressive decline is almost identical to the increase in the overall level of annual use of mechanized harvesting. When evaluating the total cost of collection, the results also depend on the specific country, where the input items for the calculation of individual fixed and variable components often change (depreciation years, insurance price, road taxes, garage price, repair costs, fuel price and costs) for live work. A very important part of the cost is represented by direct costs (price of the machine, set) and indirect costs that arise from losses. The results obtained by us point to acceptable losses that correspond to the results of other authors in the deployment of mechanized collection [12–15,33,37].

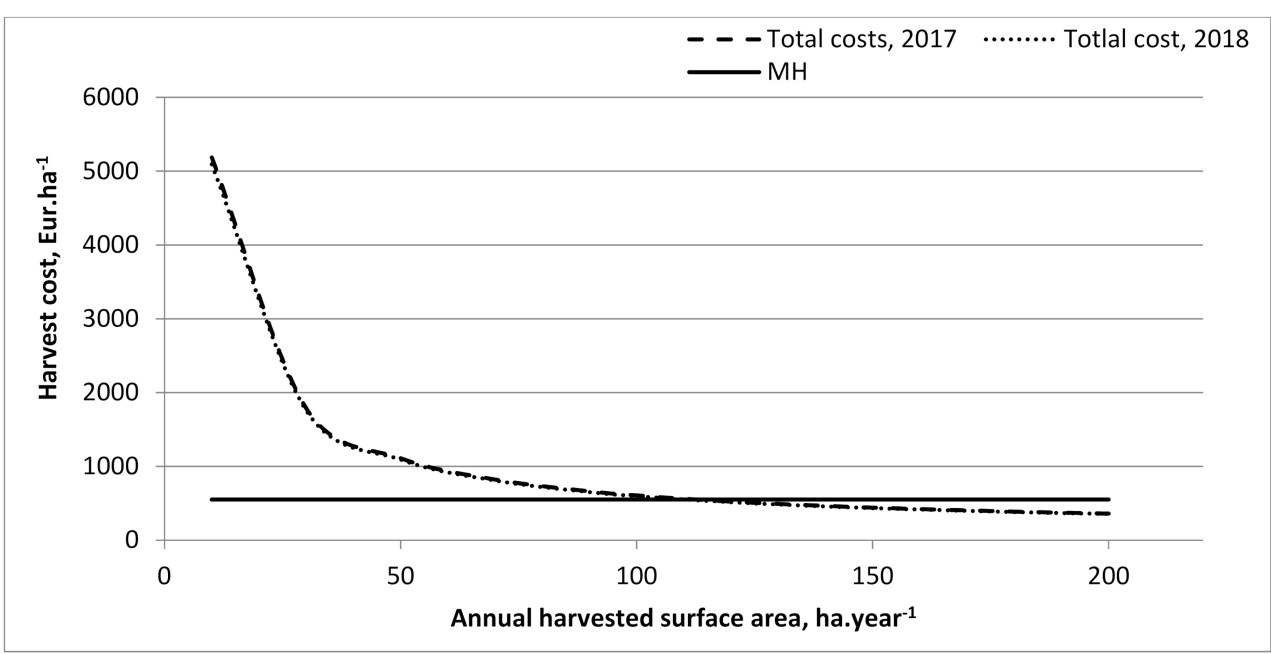

**Figure 4.** Harvesting costs with harvested area for total vineyards, year 2017 and 2018. MH: Manual harvesting.

### 3.4. Break-Even Analysis

For other possibilities of evaluation of the deployment of mechanized harvesting, our attention was focused on changing the input parameters such as the price of grapes and the cost of live work. The price value of a grape product may vary from country to country. The change often occurs with a change in the year of collection, respectively. it is also given by variety. That is why we focused on evaluating the efficiency of using a machine set for grape harvest by determining the dependence of the required minimum area of the vine on the price of grapes (price change from 50 to 500 Euro·ha$^{-1}$) The results show an exponential increase in the required input area depending on the price of grapes when considering other input cost items (Figure 5). For smaller areas, the price of grapes would have to be too undersized. On average, depending on the country and variety, grape prices are min. from 400 Euro·t$^{-1}$ (prices valid for small consumption from small winegrowers). The minimum areas of effective deployment of mechanized harvesting would be 156.1 ha for 2017 and 137.3 ha for 2018. The value is calculated by comparing the results of manual harvesting costs and machine harvesting costs. The costs of manual collection will be determined on the basis of hourly wage and time required for collection expressed in h·ha$^{-1}$. No grape losses are considered for manual harvesting. The costs of machine harvesting consist of the total unit costs added to the cost of the lost grapes (we change the price for the grapes and look for the area in which the costs of manual and machine harvesting equalize the losses).

Another monitored essential and variable parameter is the price of work, which we used to evaluate the effectiveness of the deployment of the machine set in the range of 2 to 14 Euro·h$^{-1}$ (Figure 6). The average hourly wage is very diverse in our conditions, given that the harvesting of grapes involves more or less work on a contract basis. Again, it is an input item that depends to a large extent on the country where the collection takes place. The trend in the results points to the fact that increasing the employee's hourly wage leads to a degressive decrease in the required minimum area. With regard to the examined years (2017 and 2018), there was no significant change, i.e., the impact of the examined years on the observed changes in wage costs, but on unchanged other values was not significant. In the calculations, the efficiency of manual harvesting was 9.8–10.3 ha·h$^{-1}$ and the price of grapes 500 Euro·t$^{-1}$.

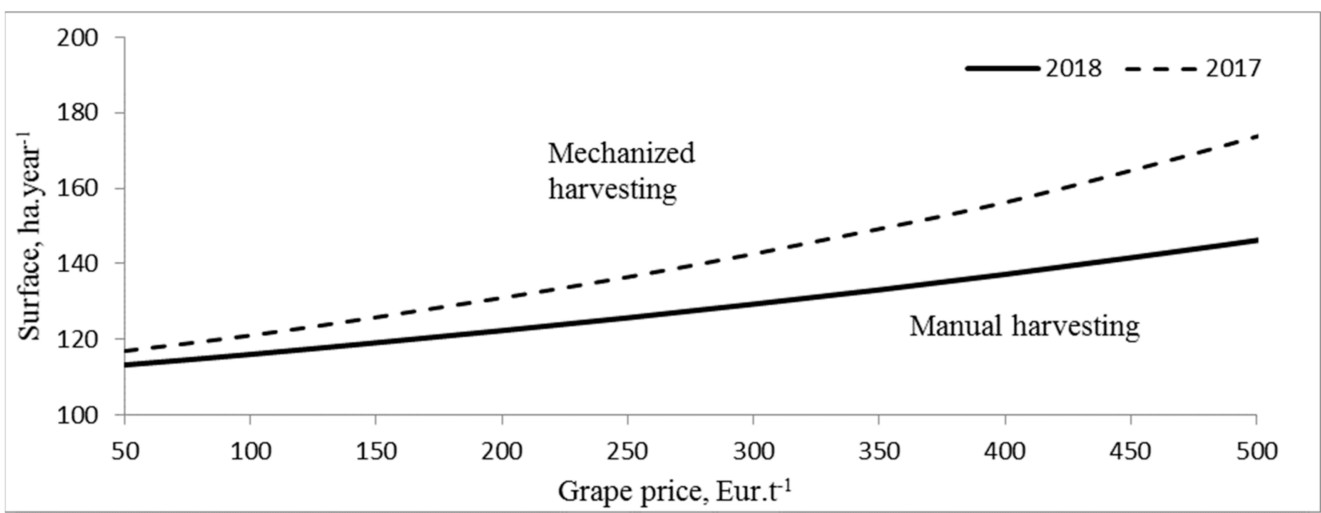

**Figure 5.** Break-even areas under different combinations of harvested area and grape price.

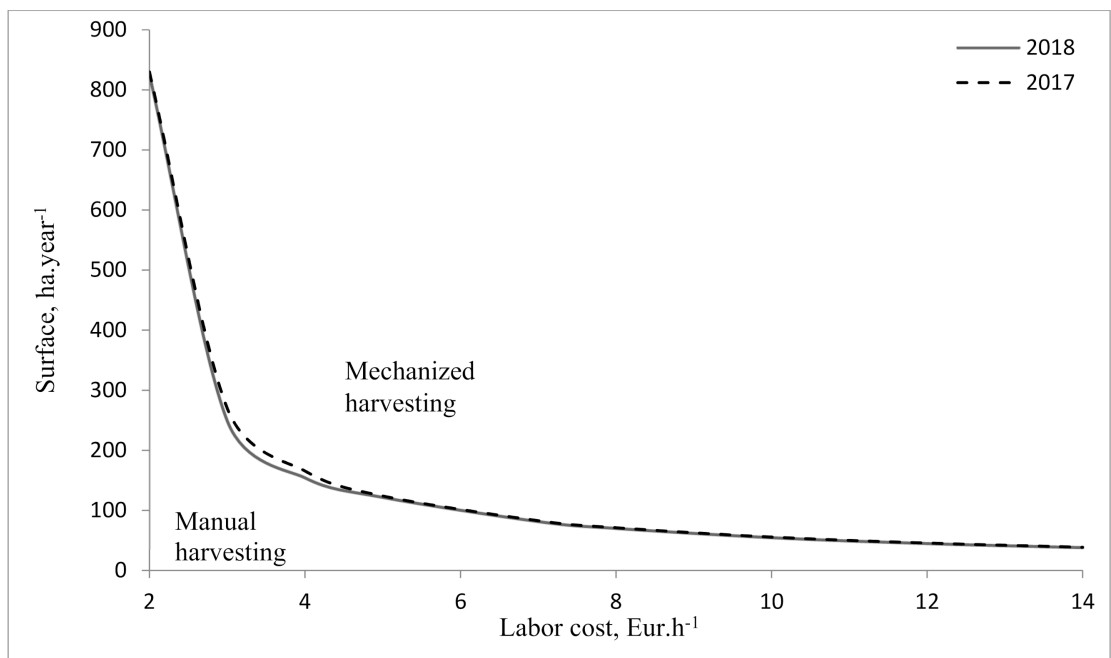

**Figure 6.** Break-even areas under different combinations of harvested area and labor costs.

When deploying the services, we would pay 13,200 Euro (a total of 26,400 Euro) for the collection of grapes with an area of 33 ha for 2017 and for 2018. If the prices did not change for 4 years, we would pay a total of 52,800 Euro for the collection. When deploying long-term depreciation, e.g., for 8 years, we would pay 105,600 Euro for collection in the form of services. The biggest problem from practical experience is getting to the order of service customers. That is why the company decided to solve the collection problem (in our case mechanized collection) by buying a machine.

The results, in contrast to the sources available so far, point to changes during several years of research and a reduction in the required minimum acreage with the use of modern and new techniques for grape harvesting. The results of not only the yield of individual varieties but also the achieved losses were evaluated statistically. The results were also subjected to statistical analysis between years. In terms of examining losses, differences were shown not only between varieties, but also years, and were statistically verified.

From the results of cost calculations for small areas of individual examined varieties, we pointed out the low efficiency of the use of machine harvesting with new technology.

With smaller areas and under the expected conditions of the service life of the machinery (tractor + harvester-new equipment), services or manual collection are economically more advantageous [35].

In the analysis of input costs, and especially variable ones, it is a complex process in which other aspects that cannot be easily quantified cannot be ruled out. The advantage of manual harvesting with qualified harvesters (they do this for several years) over mechanized is a less damaged product, which also leads to less input problems in the subsequent process of receiving and further processing of grapes for wine production. The advantage of mechanized collection over manual is the possibility of using technology with one or with two persons, while in the case of manual collection, due to the overall size, it requires a variety of low-skilled persons for collection, in the labor shortages as examined above [38].

Similar views on the effectiveness of the mechanized grape harvesting set and the lack of manpower for manual harvesting are shared by the findings from other authors [12,13,39–41].

## 4. Conclusions

The paper deals with the evaluation of the efficiency of the use of the (towed) trailed grape harvester in the conditions of Slovak viticulture, both from a technical point of view and from an economic point of view. Technical evaluation was the monitoring of post-harvest losses caused by non-harvesting or falling into the soil. Harvest losses in 2017 reached the maximum value in the examined Tramin Red variety. However, by changing the shaking parameters (number of oscillations), a reduction in losses was achieved. Therefore, it can be said that by monitoring the process of collection and the occurrence of losses by non-harvesting or overflow, it is possible to achieve a higher quality of work with the function of setting and fine-tuning. However, we must not forget the fact that the quality of harvest and thus losses are also affected by inputs such as the ripeness of the grapes (however, we did not address the issue in the article) and compliance with agrotechnical requirements.

The evaluation of mechanized collection can be considered effective only if lower unit direct costs than the available market unit price of labor (in the form of services) are achieved after all input costs for a specific area have been included. Compared to manual collection, the problem arises rather in the process of recruiting employees for this type of work. The performance of the machine set is also in our case limited by the capacity of other equipment related to other work operations of further processing of grapes (grape reception, pressing plant, etc.). Therefore, even in the given company it is a gradual harvesting of individual varieties in connection with further processing.

After the economic evaluation of mechanized collection, where the costs of operating the machine consist of fixed and variable items (the largest of which in the new technology is the purchase price), the minimum area for effective deployment was determined. Reducing the recalculated unit costs will only be achieved by increasing the acreage, which is also an example of our evaluation of research. From the results, it is possible to conclude that in the conditions of Slovak viticulture, even in small areas, it is currently the most appropriate solution to purchase harvesting equipment from several companies or to perform services in addition to harvesting their own vineyards. As the four-year depreciation rate is valid for the machine set, the determined efficiency of the technology deployment is only when managing an area of over 100 ha. From a practical point of view, however, the deployment of technology at low acreage achieves more annual use than the depreciation period, which in turn implies that the purchase price would be calculated over several years and unit costs would be reduced.

However, with an increase in the depreciation period from 4 years to 6 years and with other unchanged parameters (hourly wage, price for grapes) the required total area would decrease to 106.41 ha and with a change in the depreciation period to 8 years this value would decrease in in 2017 on 86.42 ha. In 2018, this area would be reduced to 94.85 ha at the set depreciation period (6 years) and to 77.31 ha at 8-year depreciation. Another

possibility to increase the usability of the harvesting kit would be the deployment in the form of services, which, however, we did not analyze in the study.

A statistical analysis demonstrated both the harvest and the other hand of the total losses arising from the harvest of the grapes; the analysis showed that the yields obtained were not statistically significant with respect to the year of harvest or grape variety. Furthermore, the statistical analysis of Anova's results of total losses in the mechanized harvesting of different varieties shows that in 2017 there were no statistically significant differences. Nevertheless, in 2018 statistically significant differences were achieved. When comparing the losses depending on the year of harvest, only one variety (Tramin Red) showed results that were statistically significant.

The results obtained during the multi-annual monitoring of mechanized harvesting on several varieties point to the advantages of using mechanized harvesting with a higher minimum area and in multi-annual use. Deployment of mechanized harvesting in three different varieties by deployment in the conditions of a small winery with a modern grape harvester has proven to be beneficial and significant.

Overall, the outputs of the present article show a tendency towards the suitability of the use of technology due to the speed and low losses during harvesting. In addition, with long-term deployment, the efficiency of machine harvesting will clearly increase. Small winegrowers are unlikely to go in this direction, but in terms of mass production, it is the only way, apart from the complex process of recruiting staff for harvesting. In the conditions of Slovak viticulture, the number of used trailed harvesters has increased in the last 10 years. The self-propelled harvester is also advancing.

**Author Contributions:** Conceptualization, J.J. and K.K.; methodology, M.D.; software, M.D.; validation, J.J., K.K. and M.D.; formal analysis, M.D.; investigation, M.D.; resources, J.J.; data curation, M.D.; writing—original draft preparation, J.J. and K.K.; writing—review and editing, K.K.; visualization, J.J.; supervision, P.B.; project administration, P.B.; funding acquisition, J.J. All authors have read and agreed to the published version of the manuscript.

**Funding:** This research was funded by Cultural and educational grant agency of the Ministry of Education, Science, Research and Sports of the Slovak Republic, grant number KEGA 012SPU-4/2020.

**Institutional Review Board Statement:** Not applicable.

**Informed Consent Statement:** Not applicable.

**Data Availability Statement:** Not applicable.

**Acknowledgments:** This publication is the result of the project implementation: Innovation of the Education Process and Implementation of Practical Knowledge Focused on the Wine Production and Viticulture (KEGA 012SPU-4/2020).

**Conflicts of Interest:** The authors declare no conflict of interest.

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
