# Peer review of "Mechanized Grape Harvest Efficiency"

_applsci, doi:10.3390/app11104621_

Round 1
Reviewer 1 Report
The article presented to me for evaluation is an interesting extension of the possibilities and profitability of the mechanical grape harvesting process. I think that as an example of this type of study, it may be interesting for people considering investments in this area. However, I have a few comments about my work.
- In Table 3, under Dimension, Length / Width / Height, add a unit.
- Instead of using rotations per minute (rpm) use the air flow velocity or the resulting overpressure parameters when determining the separating fans (can be measured on an actual model). This way the parameter will be clearer for interpreting the results. I propose to consider it in the following lines: 137, 257 and 283.
- In the case of formula (2) in row 174, however, suggests considering the depreciation costs of the grape harvester.
- In table 4, under Machine dimensions, change the unit or value.
- On line 252, change the decimal point.
- In line 255, please correct the values. An editorial error has probably occurred. As it stands, the values are absurd.
- In table 5 for Field efficiency, Ef and Effective field capacity, Ca please do not add SE (it can be confusing).
- In Figures 4 and 5, remove the decimal values from the ordinate axes (Harvest costs and Surface).
- Proposes to present more detailed (as calculated) the minimum area of mechanical set for which it is effective lines 415-416.
- There are no more precise conclusions in the conclusions (and previous analyzes), calculating the profitability on a smaller area of using a mechanical set, eg with a longer depreciation period, or maybe, for example, additionally taking into account the further resale of the set as used to another buyer.
- An interesting analysis could be pledging a mechanical grape harvest in a smaller area and additionally carrying it out to other farmers as a service (what size this service would have to be profitable).
- In the study, no typical statistical analysis was actually carried out (or at least not confirmed by the results and analyzes), only SE was determined. I do not see any determination of the significance of the differences anywhere, and there is such information in lines 164-162 and 248-249. It would be worth putting them in your work or modifying this informa.
I believe that the work after modifications is suitable for publication.
Greetings
Author Response
Thank you for precise review. Please see the attachment.

Reviewer 2 Report
Very important article. The cultivation of grapes is one of the difficult ones due to changing weather conditions, selection of suitable soils or proper irrigation.
The mechanization of grape harvesting is an important aspect in vineyard cultivation. This is all the more important in the case of a small country like the Slovak Republic (large vineyards on the one hand and small domestic ones on the other).
I have a few comments:
- in the article, or more precisely in the research methodology, the time interval in which the research was done is missing (was it annual or maybe biennial research?),
- no meteorological data,
- statistical analysis could be clearer.
Author Response

(The authors gave the same response as above.)

Reviewer 3 Report
I recommend the authors for a shorter abstract.
Introduction section is insufficient to prove the state of the art in the area. What’s the actual technical offer in viticulture? Which are the boundaries? What’s the human resources offer in the field? What is the human resources market situation in Slovakia (any statistical data, something official)? Did any other authors study the actual comparison between manual and technical work in the field? What are their results? What are the authors doing different and what bring their study new in the field?
Rows 69-70 “viticulture and viticulture”?!
Pag. 3: where from does the authors have the information about Pinot Gris variety, Pinot Noir variety, Tramin red variety? They need to mention the cited sources.
With all due respect for the authors, the technical characteristics of 2 standard tractors (covering 1 page of the paper, and no cited sources) aren’t scientific information for the readers. So, I suggest the authors to shorten this information.
Row 323: “It is clear from the available sources...”, Row 331: “Some internet sources” can’t be scientific sources to operate at this level.
Rows 323-339 – There are too many general statements (not just here, but in the entire paper), that are dangerous and vulnerable for sustaining the general objective of the article.
Row 364: “The results resulted...”?!
Rows 364-366: How did the authors get to that number?
Apologize to the authors, but I do not understand how they get the Fig. 3-6 (I couldn’t see the data...) and I can’t see the place where Fig. 5 is mentioned in the paper.
Row 447: “findings and findings...”?!
In term of “the value of the frequency of oscillations”, I couldn’t get the optimal value for the least losses (only a better situation, by reducing the value with 20min).
I recommend the authors to not use syllable breakdown in the paper.
In the present form, the paper is a use-case situation and looks like a cost benefit analysis (which is incomplete) to sustain the acquisition of a harvester, rather than a scientific study. Maybe the authors should improve their paper to better drive to a research paper.
Author Response

(The authors gave the same response as above.)

Reviewer 4 Report
The present document entitled “Mechanized grape harvest efficiency” presents a focus on the effectiveness of the use of an outboard grape harvester in the conditions of Slovak viticulture, also on the losses that arise during it.
The document is interesting, but in this reviewer opinion, some comments are proposed to authors.
Abstract. The abstract is well presented, describing the main achievements of the work. Nevertheless, in this reviewer opinion, the first plural person “we” should not be included in scientific text, but impersonal forms. Apart from this, the main novelty should be indicated in the abstract, inviting readers to get deeper in the main results of the research.
Introduction. This section presents the main objectives of the work. In this reviewer opinion, some more references could be added in the introduction, to present readers the importance of the scientific treatment of this problem and the importance of the researched topic in the recent past.
- Materials and Methods.
2.1 Terrain conditions. This section describes the terrain and maps of the studied region
2.2 Mechanization for grape harvest. Information about the performed experiments is provided
2.3 Combine harvester performance. In this case, many details of the methodology are provided. Again in this reviewer opinion “we” should not be the normal way to describe methodology in these sort of texts. In this reviewer opinion, more information about economic assessment, with more technical base would be welcome in this section.
- Results and discussion
This is a section describing results of the previous described methodology for harvesting. It is interesting to see that the results of similar studies have been documented in previous researches, but in this reviewer opinion, these similar studies should be pointed out also in the introduction, to document previous similar works, and pointing out the differences with these previous studies.
In this section, economic results are well documented, and the results clearly presented in the figures. But some question arises from the reading of the final comment in the results and discussion section (line 446) “Similar views on the effectiveness of the mechanized grape harvesting set and the lack of manpower for manual harvesting are shared by the findings and findings from other authors”. Which is then the main contribution of this research in terms of results?
Where is section 4?
- Conclusions. Conclusions section presents a resume of the previous Research. In this reviewer opinion, the novelty and main contributions of the work should be clearly pointed out in the conclusions.
Author Response

(The authors gave the same response as above.)

Round 2
Reviewer 3 Report
Thanks for the additions made by the authors, now the article is more suitable for publication. Good luck!
A minor notice:
“The average territorial air temperature in Slovakia was 8.4 ° C” (lines 146-147) vs “The average territorial air temperature in Slovakia was 8.6 ° C” (line 150) ?!
Reviewer 4 Report
In this reviewer opinion, authors have implemented correctly the previous suggested observations.